# Counterfactual Predictions under Runtime Confounding

**Amanda Coston**
Heinz College & Machine Learning Department
Carnegie Mellon University
acoston@cs.cmu.edu

**Edward H. Kennedy**
Department of Statistics
Carnegie Mellon University
edward@stat.cmu.edu

**Alexandra Chouldechova**
Heinz College
Carnegie Mellon University
achould@cmu.edu

## Abstract

Algorithms are commonly used to predict outcomes under a particular decision or intervention, such as predicting likelihood of default if a loan is approved. Generally, to learn such *counterfactual* prediction models from observational data on historical decisions and corresponding outcomes, one must measure all factors that jointly affect the outcome and the decision taken. Motivated by decision support applications, we study the counterfactual prediction task in the setting where all relevant factors are captured in the historical data, but it is infeasible, undesirable, or impermissible to use some such factors in the prediction model. We refer to this setting as **runtime confounding**. We propose a doubly-robust procedure for learning counterfactual prediction models in this setting. Our theoretical analysis and experimental results suggest that our method often outperforms competing approaches. We also present a validation procedure for evaluating the performance of counterfactual prediction methods.

## 1 Introduction

Algorithmic tools are increasingly prevalent in domains such as health care, education, lending, criminal justice, and child welfare [4, 38, 18, 15, 9]. In many cases, the tools are not intended to replace human decision-making, but rather to distill rich case information into a simpler form, such as a risk score, to inform human decision makers [3, 11]. The type of information that these tools need to convey is often *counterfactual* in nature. Decision-makers need to know what is likely to happen if they choose to take a particular action. For instance, an undergraduate program advisor determining which students to recommend for a personalized case management program might wish to know the likelihood that a given student will graduate if enrolled in the program. In child welfare, case workers and their supervisors may wish to know the likelihood of positive outcomes for a family under different possible types of supportive service offerings.

A common challenge to developing valid counterfactual prediction models is that all the data available for training and evaluation is observational: the data reflects historical decisions and outcomes under those decisions rather than randomized trials intended to assess outcomes under different policies. If the data is confounded—that is, if there are factors not captured in the data that influenced both the outcome of interest and historical decisions—valid counterfactual prediction may not be possible. In this paper we consider the setting where all relevant factors are captured in the data, and so historical

decisions and outcomes are unconfounded, but where it is infeasible, undesirable, or impermissible to use some such factors in the prediction model. We refer to this setting as **runtime confounding**.

Runtime confounding naturally arises in a number of different settings. First, relevant factors may not yet be available at the desired runtime. For instance, in child welfare screening, call workers decide which allegations coming in to the child abuse hotline should be investigated based on the information in the call and historical administrative data [9]. The call worker's decision-making process can be informed by a risk assessment if the call worker can access the risk score in real-time. Since existing case management software cannot run speech/NLP models in realtime, the call information (although recorded) is not available at runtime, thereby leading to runtime confounding. Second, runtime confounding arises when historical decisions and outcomes have been affected by sensitive or protected attributes which for legal or ethical reasons are deemed ineligible as inputs to algorithmic predictions. We may for instance be concerned that call workers implicitly relied on race in their decisions, but it would not be permissible to include race as a model input. Third, runtime confounding may result from interpretability or simplicity requirements. For example, a university may require algorithmic tools used for case management to be interpretable. While information conveyed during student-advisor meetings is likely informative both of case management decisions and student outcomes, natural language processing models are not classically interpretable, and thus the university may wish instead to only use structured information like GPA in their tools.

In practice, when it is undesirable or impermissible to use particular features as model inputs at runtime, it is common to discard the ineligible features from the training process. This can induce considerable bias in the resulting prediction model when the discarded features are significant confounders. To our knowledge, the problem of learning valid counterfactual prediction models under runtime confounding has not been considered in the prior literature, leaving practitioners without the tools to properly incorporate runtime-ineligible confounding features into the training process.

**Contributions:** Drawing upon techniques used in low-dimensional treatment effect estimation [46, 52, 8], we propose a procedure for the full pipeline of learning and evaluating prediction models under runtime confounding. We (1) formalize the problem of counterfactual prediction with runtime confounding [§ 2]; (2) propose a solution based on doubly-robust techniques that has desirable theoretical properties [§ 3.3]; (3) theoretically and empirically compare this solution to an alternative counterfactually valid approach as well as the standard practice, describing the conditions under which we expect each to perform well [§ 3 & 5]; and (4) provide an evaluation procedure to assess performance of the methods in the real-world [§ 4]. Proofs, code and results of additional experiments are presented in the Supplement.

## 1.1 Related work

Our work builds upon a growing literature on counterfactual risk assessments for decision support that proposes methods for the unconfounded prediction setting [35, 10]. Following this literature, our goal is to predict outcomes under a proposed decision (interchageably referred to as 'treatment' or 'intervention') in order to inform human decision-makers about what is likely to happen under that treatment. This prediction task is different from the common causal inference problem of treatment effect estimation, which targets a contrast of outcomes under two different treatments [48, 37]. Treatment effects are useful for describing responsiveness to treatment. While responsiveness is relevant to some types of decisions, it is insufficient, or even irrelevant, to consider for others. For instance, a doctor considering an invasive procedure may make a different recommendation for two patients with the same responsiveness if one has a good probability of successful recovery without the procedure and the other does not. In lending settings, the responsiveness to different loan terms is irrelevant; all that matters is that the likelihood of default be sufficiently small under feasible terms.

Our proposed prediction (Contribution 2) and evaluation methods (Contribution 4) draw upon the literature on double machine learning and doubly-robust estimation, which uses the efficient influence function to produce estimators with reduced bias [46, 30, 29, 16, 7]. Of particular relevance are methods for estimating treatment effects conditional on only a subset of confounders [46, 52, 8, 36, 25] and for learning treatment assignment policies conditional on only a subset of confounders [50, 2, 19]. In our case, we are interested in *predictions* conditional on only those features that are permissible or desirable to consider at runtime. Our methods are specifically designed for minimizing prediction error, rather than providing inferential guarantees such as confidence intervals, as is common in the treatment effect estimation setting.

Our work is also related to the literature on marginal structure models (MSMs) [31, 28]. An MSM is a model for a marginal mean of a counterfactual, possibly conditional on a subset of baseline covariates. The standard MSM approach is semiparametric, employing parametric assumptions for the marginal mean but leaving other components of the data-generating process unspecified [45]. Nonparametric variants were studied in the unconditional case for continuous treatments by Rubin and van der Laan [32]. In contrast our setting can be viewed as a nonparametric MSM for a binary treatment, conditional on a large subset of covariates. This is similar in spirit to partly-conditional treatment effect estimation [44]; however we do not target a contrast since our interest is in predictions rather than treatment effects. Our results are also less focused on model selection [43], and more on error rates for particular estimators. We draw on techniques for sample-splitting and cross-fitting, which have been used in the regression setting for model selection and tuning [13, 46] and in treatment effect estimation [27, 51, 6].

Our method is relevant to settings where the outcome is selectively observed. This *selective labels* problem [21, 20] is common in settings like lending where the repayment/default outcome is only observed for applicants whose loan is approved. Runtime confounding can arise in such settings if some factors that are used for decision-making are unavailable for prediction.

Recent work has considered methods to accommodate confounding due to sources other than missing confounders at runtime. A line of work has considered how to use causal techniques to correct runtime dataset shift [40, 24, 39]. In our case the runtime setting is different from the training setting not because of distributional shift but because we can no longer access all confounders. These methods also differ from ours in that they are not seeking to predict outcomes under specific decisions.

There is also a line of work that considers confounding in the *training* data [14, 23]. While confounded training data is common in various applications, our work targets decision support settings where the factors used by decision-makers are recorded in the training data but are not available for prediction.

Lastly, there are connections between runtime confounding and the literature on privileged learning and algorithmic fairness that use features during training time that are not available for prediction. Learning using Privileged Information (LUPI) has been proposed for settings in which the training data contains additional features that are not available at runtime [47]. In algorithmic fairness, disparate learning processes (DLPs) use the sensitive attribute during training to produce models that achieve a target notion of parity without requiring access to the protected attribute at test time [22]. LUPI and DLPs both make use of variables that are only available at train time, but if these variables affect the decisions under which outcomes are observed, predictions from LUPI and DLPs will be confounded because neither accounts for how these variables affect decisions. By contrast, our method uses confounding variables during training to produce valid counterfactual predictions.

## 2 Problem setting

Our goal is to predict outcomes under a proposed treatment $A = a \in \{0, 1\}$ based on runtime-available predictors $V \in \mathcal{V} \subseteq \mathbb{R}^{d_V}$.[1] Using the potential outcomes framework [33, 26], our prediction target is $\nu_a(v) := \mathbb{E}[Y^a \mid V = v]$ where $Y^a \in \mathcal{Y} \subseteq \mathbb{R}$ is the potential outcome we would observe under treatment $A = a$. We let $Z \in \mathcal{Z} \subseteq \mathbb{R}^{d_Z}$ denote the runtime-hidden confounders, and we denote the propensity to receive treatment $a$ by $\pi_a(v, z) := \mathbb{P}(A = a \mid V = v, Z = z)$. We also define the outcome regression by $\mu_a(v, z) := \mathbb{E}[Y^a \mid V = v, Z = z]$. For brevity, we will generally omit the subscript, using notation $\nu$, $\pi$ and $\mu$ to denote the functions for a generic treatment $a$.

**Definition 2.1.** Formally, the task of counterfactual prediction under **runtime-only confounding** is to estimate $\nu(v)$ from iid training data $(V, Z, A, Y)$ under the following two conditions:

**Condition 2.1.1** (Training Ignorability)**.** Decisions are unconfounded given $V$ and $Z$: $Y^a \perp A \mid V, Z$.

**Condition 2.1.2** (Runtime Confounding)**.** Decisions are confounded given only $V$: $Y^a \not\perp A \mid V$; equivalently, $A \not\perp Z \mid V$ and $Y^a \not\perp Z \mid V$

To ensure that the target quantity is identifiable, we require two further assumptions, which are standard in causal inference and not specific to the runtime confounding setting.

**Algorithm 1** The plug-in (PL) approach

---

*Stage 1:* Learn $\hat{\mu}(v, z)$ by regressing $Y \sim V, Z \mid A = a$
*Stage 2:* Learn $\hat{\nu}_{\mathrm{PL}}(v)$ by regressing $\hat{\mu}(V, Z) \sim V$

---

**Condition 2.1.3** (Consistency). A case that receives treatment $a$ has outcome $Y = Y^a$.

**Condition 2.1.4** (Positivity). $\mathbb{P}(\pi_a(V, Z) \geq \epsilon > 0) = 1 \quad \forall a$

**Identifications.** Under conditions 2.1.1-2.1.4, we can write the counterfactual regression functions $\mu$ and $\nu$ in terms of observable quantities. We can identify $\mu(v, z) = \mathbb{E}[Y \mid V = v, Z = z, A = a]$ and our target $\nu(v) = \mathbb{E}[\mathbb{E}[Y \mid V = v, Z = z, A = a] \mid V = v] = \mathbb{E}[\mu(V, Z) \mid V = v]$. The iterated expectation in the identification of $\nu$ suggests a two-stage approach that we propose in § 3.2 after reviewing current approaches.

**Miscellaneous notation.** Throughout the paper we let $p(x)$ denote probability density functions; $\hat{f}$ denote an estimate of $f$; $L \lesssim R$ indicate that $L \leq C \cdot R$ for some universal constant $C$; $\mathbb{I}$ denote the indicator function; and define $\|f\|^2 := \int (f(x))^2 p(x) dx$.

## 3 Prediction methods

### 3.1 Standard practice: Treatment-conditional regression (TCR)

Standard counterfactual prediction methods train models on the cases that received treatment $a$ [35, 10], a procedure we will refer to as **treatment-conditional regression** (TCR). This procedure estimates $\omega(v) = \mathbb{E}[Y \mid A = a, V = v]$. This method works well given access to all the confounders at runtime; if $A \perp Y^a \mid V$, then $\omega(v) = \mathbb{E}[Y^a \mid V = v] = \nu(v)$. However, under runtime confounding, $\omega(v) \neq \mathbb{E}[Y^a \mid V = v]$, so this method does not target the right counterfactual quantity, and may produce misleading predictions.[2] For instance, consider a risk assessment setting that historically assigned risk-mitigating treatment to cases that have higher risk under the null treatment ($A = 0$). Using TCR to predict outcomes under the null treatment will underestimate risk since $\mathbb{E}[Y \mid V, A = 0] = \mathbb{E}[Y^0 \mid V, A = 0] < \mathbb{E}[Y^0 \mid V]$. We can characterize the bias of this approach by analyzing $b(v) := \omega(v) - \nu(v)$, a quantity we term the pointwise *confounding bias*.

**Proposition 3.1.** Under runtime confounding, $\omega(v)$ has pointwise confounding bias

$$b(v) = \int_{\mathcal{Z}} \mu(v, z) \Big( p(z \mid V = v, A = a) - p(z \mid V = v) \Big) dz \quad \neq \quad 0 \tag{1}$$

By Condition 2.1.2, this confounding bias will be non-zero. Nonetheless we might expect the TCR method to perform well if $b(v)$ is small enough. We can formalize this intuition by decomposing the error of a TCR predictive model $\hat{\nu}_{\mathrm{TCR}}$ into estimation error and confounding bias:

**Proposition 3.2.** The pointwise regression error of the TCR method can be bounded as follows:

$$\mathbb{E}[(\nu(v) - \hat{\nu}_{\mathrm{TCR}}(v))^2] \lesssim \mathbb{E}[(\omega(v) - \hat{\nu}_{\mathrm{TCR}}(v))^2] + b(v)^2$$

### 3.2 A simple proposal: Plug-in (PL) approach

We can avoid the confounding bias of TCR through a simple two-stage procedure we call the **plug-in** approach that targets the proper counterfactual quantity. This approach, described in Algorithm 1, first estimates $\mu$ and then uses $\mu$ to construct a pseudo-outcome which is regressed on $V$ to yield prediction $\hat{\nu}_{\mathrm{PL}}$. Cross-fitting techniques (Alg. 2) can be applied to prevent issues that may arise due to potential overfitting when learning both $\hat{\mu}$ and $\hat{\nu}_{\mathrm{PL}}$ on the same training data. Sample-splitting (or cross-fitting) also enables us to get the following upper bound on the error of the PL approach.

**Proposition 3.3.** Under sample-splitting for stages 1 and 2 and stability conditions on the 2nd stage estimators (appendix, [17]), the PL method has pointwise regression error bounded by

$$\mathbb{E}\Big[ \big( \hat{\nu}_{\mathrm{PL}}(v) - \nu(v) \big)^2 \Big] \lesssim \mathbb{E}\Big[ \big( \tilde{\nu}(v) - \nu(v) \big)^2 \Big] + \mathbb{E}\Big[ \big( \hat{\mu}(V, Z) - \mu(V, Z) \big)^2 \mid V = v \Big]$$

**Algorithm 2** The plug-in (PL) approach with cross-fitting

---

Randomly divide training data into two partitions $\mathcal{W}^1$ and $\mathcal{W}^2$.
**for** $(p, q) \in \{(1, 2), (2, 1)\}$ **do**
    *Stage 1:* On partition $\mathcal{W}^p$, learn $\hat{\mu}^p(v, z)$ by regressing $Y \sim V, Z \mid A = a$
    *Stage 2:* On partition $\mathcal{W}^q$, learn $\hat{\nu}_{\text{PL}}^q(v)$ by regressing $\hat{\mu}^p(V, Z) \sim V$
**PL prediction:** $\hat{\nu}_{\text{PL}}(v) = \frac{1}{2} \sum_{i=1}^{2} \hat{\nu}_{\text{PL}}^i(v)$

---

---

**Algorithm 3** The proposed doubly-robust (DR) approach

---

*Stage 1:* Learn $\hat{\mu}(v, z)$ by regressing $Y \sim V, Z \mid A = a$.
    Learn $\hat{\pi}(v, z)$ by regressing $\mathbb{I}\{A = a\} \sim V, Z$
*Stage 2:* Learn $\hat{\nu}_{\text{DR}}(v)$ by regressing $\left( \frac{\mathbb{I}\{A=a\}}{\hat{\pi}(V,Z)} (Y - \hat{\mu}(V, Z)) + \hat{\mu}(V, Z) \right) \sim V$

---

where the oracle-quantity $\tilde{\nu}(v)$ describes the function we would get in the second-stage if we had oracle access to $Y^a$.

This simple approach can consistently estimate our target $\nu(v)$. However, it solves a harder problem (estimation of $\mu(v, z)$) than what our lower-dimensional target $\nu$ requires. Notably the bound depends *linearly* on the MSE of $\hat{\mu}$. We next propose an approach that avoids such strong dependence.

### 3.3 Our main proposal: Doubly-robust (DR) approach

Our main proposed method is what we call the **doubly-robust** (DR) approach, which improves upon the PL procedure by using a bias-corrected pseudo-outcome in the second stage (Alg. 4). The DR approach estimates both $\mu$ and $\pi$, which enables the method to perform well in situations in which $\pi$ is easier to estimate than $\mu$. We propose a cross-fitting (Alg. 3) variant that satisfies the sample-splitting requirements of Theorem 3.1.

**Theorem 3.1.** Under sample-splitting to learn $\hat{\mu}$, $\hat{\pi}$, and $\hat{\nu}_{\text{DR}}$ and stability conditions on the 2nd stage estimators (appendix, [17]), the DR method has pointwise error bounded by:

$$\mathbb{E}\left[ \left( \hat{\nu}_{\text{DR}}(v) - \nu(v) \right)^2 \right] \lesssim \mathbb{E}\left[ \left( \tilde{\nu}(v) - \nu(v) \right)^2 \right]$$
$$+ \mathbb{E}\left[ (\hat{\pi}(V, Z) - \pi(V, Z))^2 \mid V = v \right] \mathbb{E}\left[ (\hat{\mu}(V, Z) - \mu(V, Z))^2 \mid V = v \right]$$

This implies a similar bound on the integrated MSE (given in appendix).

The DR error is bounded by the error of an oracle with access to $Y^a$ and a *product* of nuisance function errors.[3] This product can be substantially smaller than the error of $\hat{\mu}$ in the PL bound. When this product is less than the oracle error, the DR approach is oracle-efficient, in the sense that it achieves (up to a constant factor) the same error rate as an oracle. This model-free result provides bounds that hold for *any* regression method. It is nonetheless instructive to consider the form of these bounds in a specific context. The next result is specialized to the sparse high-dimensional setting.[4]

**Corollary 3.1.** Assume stability conditions on the 2nd stage regression estimator (appendix, [17]) and that a $k$-sparse model can be estimated with squared error $k^2 \sqrt{\frac{\log d}{n}}$ (e.g. [5]). With $k_\omega$-sparse $\omega$, the pointwise error for the TCR method is

$$\mathbb{E}\left[ \left( \hat{\nu}_{\text{TCR}}(v) - \nu(v) \right)^2 \right] \lesssim k_\omega^2 \sqrt{\frac{\log d_{\text{V}}}{n}} + b(v)^2$$

With $k_\mu$-sparse $\mu$ and $k_\nu$-sparse $\nu$, the pointwise error for the PL method is

$$\mathbb{E}\left[ \left( \hat{\nu}_{\text{PL}}(v) - \nu(v) \right)^2 \right] \lesssim k_\nu^2 \sqrt{\frac{\log d_{\text{V}}}{n}} + k_\mu^2 \sqrt{\frac{\log d}{n}}$$

Additionally with $k_\pi$-sparse $\pi$, the pointwise error for the DR method is

$$\mathbb{E}\left[ \left( \hat{\nu}_{\text{DR}}(v) - \nu(v) \right)^2 \right] \lesssim k_\nu^2 \sqrt{\frac{\log d_{\text{V}}}{n}} + k_\mu^2 k_\pi^2 \frac{\log d}{n}$$

**Algorithm 4** The proposed doubly-robust (DR) approach with cross fitting

---

Randomly divide training data into three partitions $\mathcal{W}^1, \mathcal{W}^2, \mathcal{W}^3$.
**for** $(p, q, r) \in \{(1, 2, 3), (3, 1, 2), (2, 3, 1)\}$ **do**
    *Stage 1:* On $\mathcal{W}^p$, learn $\hat{\mu}^p(v, z)$ by regressing $Y \sim V, Z \mid A = a$.
        On $\mathcal{W}^q$, learn $\hat{\pi}^q(v, z)$ by regressing $\mathbb{I}\{A = a\} \sim V, Z$
    *Stage 2:* On $\mathcal{W}^r$, learn $\hat{\nu}^r_{\mathrm{DR}}$ by regressing $\left( \frac{\mathbb{I}\{A=a\}}{\hat{\pi}^q(V,Z)}(Y - \hat{\mu}^p(V, Z)) + \hat{\mu}^p(V, Z) \right) \sim V$
**DR prediction:** $\hat{\nu}_{\mathrm{DR}}(v) = \frac{1}{3} \sum_{i=1}^3 \hat{\nu}^i_{\mathrm{DR}}(v)$

---

---

**Algorithm 5** Cross-fitting approach to evaluation of counterfactual prediction methods

---

**Input:** Test samples $\{(V_j, Z_j, A_j, Y_j)\}_{j=1}^{2n}$ and prediction models $\{\hat{\nu}_1, \ldots \hat{\nu}_h\}$
Randomly divide test data into two partitions $\mathcal{W}^0 = \{(V_j^0, Z_j^0, A_j^0, Y_j^0)\}_{j=1}^n$ and $\mathcal{W}^1 = \{(V_j^1, Z_j^1, A_j^1, Y_j^1)\}_{j=1}^n$.
**for** $(p, q) \in \{(0, 1), (1, 0)\}$ **do**
    On $\mathcal{W}^p$, learn $\hat{\pi}^p(v, z)$ by regressing $\mathbb{I}\{A = a\} \sim V, Z$.
    **for** $m \in \{1, \ldots, h\}$ **do**
        On $\mathcal{W}^p$, learn $\hat{\eta}^p_m(v, z)$ by regressing $(Y - \hat{\nu}_m(V))^2 \sim V, Z \mid A = a$
    On $\mathcal{W}^q$, for $j \in \{1, \ldots, n\}$ compute $\phi^q_{m,j} = \frac{\mathbb{I}\{A_j^q = a\}}{\hat{\pi}^p(V_j^q, Z_j^q)} ((Y_j^q - \hat{\nu}_m(V_j^q))^2 - \hat{\eta}^p_m(V_j^q, Z_j^q)) + \hat{\eta}^p_m(V_j^q, Z_j^q)$
**Output error estimate confidence intervals:** for $m \in \{1, \ldots, h\}$:
$$\mathrm{MSE}_m = \left( \frac{1}{2n} \sum_{i=0}^1 \sum_{j=1}^n \phi^i_{m,j} \right) \pm 1.96 \sqrt{\frac{1}{2n} \mathrm{var}(\phi_m)}$$

---

The DR approach is therefore oracle efficient when $\left( \frac{k_\mu k_\pi}{k_\nu} \right)^2 \lesssim \left( \frac{\sqrt{n} \log d_V}{\log d} \right)$.

Note that the PL approach cannot achieve oracle efficiency because $k_\mu > k_\nu$ and $d_v < d$. For exposition, consider the simple case where $k_\nu \approx k_\mu \approx k_\pi$. Corollary 3.1 indicates that when $d_V \approx d$, the DR and PL methods will perform similarly. When $d_V \ll d$, we expect the DR to outperform the PL method because the second term of the PL bound dominates the error whereas the first term of the DR bound dominates in high-dimensional settings. When $d_V \ll d$ and the amount of confounding is small, we expect the TCR to perform well. This theoretical analysis helps us understand when we expect the prediction methods to perform well. However, in practice, these upper bounds may not be tight and the degree of confounding is typically unknown. To compare the prediction methods in practice, we require a method for counterfactual model evaluation.

## 4 Evaluation method

We describe an approach for evaluating the prediction methods using observed data. In our problem setting (§ 2.1), the prediction error of a model $\hat{\nu}$ is identified as $\mathbb{E}[(Y^a - \hat{\nu}(V))^2] = \mathbb{E}[\mathbb{E}[(Y - \hat{\nu}(V))^2 \mid V, Z, A = a]]$. We propose a doubly-robust procedure to estimate the prediction error that follows the approach in [10], which focused on classification metrics and therefore did not consider MSE. Defining the error regression $\eta(v, z) := \mathbb{E}[(Y^a - \hat{\nu}(V))^2 | V = v, Z = z]$, which is identified as $\mathbb{E}[(Y - \hat{\nu}(V))^2 \mid V = v, Z = z, A = a]$, the **doubly-robust estimate of the MSE of** $\nu$ is

$$\frac{1}{n} \sum_{i=1}^n \left[ \frac{\mathbb{I}\{A_i = a\}}{\hat{\pi}(V_i, Z_i)} \left( (Y_i - \hat{\nu}(V_i))^2 - \hat{\eta}(V_i, Z_i) \right) + \hat{\eta}(V_i, Z_i) \right]$$

The doubly-robust estimation of MSE is $\sqrt{n}$-consistent under sample-splitting and $n^{1/4}$ convergence in the nuisance function error terms, enabling us to get estimates with confidence intervals. Algorithm 5 describes this procedure.[5] This evaluation method can also be used to select the regression estimators for the first and second stages.

## 5 Experiments

We evaluate our methods against ground truth by performing experiments on simulated data, where we can vary the amount of confounding in order to assess the effect on predictive performance. While

our theoretical results for PL and DR are obtained under sample splitting, in practice there may be a reluctance to perform sample splitting in training predictive models due to the potential loss in efficiency. In this section we present results where we use the full training data to learn the 1st-stage nuisance functions and 2nd-stage regressions for DR and PL and we use the full training data for the one-stage TCR.[6] This allows us to examine performance in a setting outside what our theory covers.

We first analyze how the methods perform in a sparse linear model. This simple setup enables us to explore how properties like correlation between $V$ and $Z$ impact performance. We simulate data as

$$V_i \sim \mathcal{N}(0,1) \qquad\qquad ; \ 1 \le i \le d_{\mathrm{V}}$$
$$Z_i \sim \mathcal{N}(\rho V_i, 1-\rho^2) \qquad ; \ 1 \le i \le d_{\mathrm{Z}}$$

$$\mu(V,Z) = \frac{k_v}{k_v + \rho k_z}\Big(\sum_{i=1}^{k_v} V_i + \sum_{i=1}^{k_z} Z_i\Big) \qquad Y^a = \mu(V,Z) + \epsilon \ ; \ \epsilon \sim \mathcal{N}\left(0, \frac{1}{2n}\|\mu(V,Z)\|_2^2\right)$$

$$\nu(V) = \frac{k_v}{k_v + \rho k_z}\Big(\sum_{i=1}^{k_v} V_i + \rho \sum_{i=1}^{k_z} V_i\Big)$$

$$\pi(V,Z) = 1 - \sigma\left(\frac{1}{\sqrt{k_v + k_z}}\Big(\sum_{i=1}^{k_v} V_i + \sum_{i=1}^{k_z} Z_i\Big)\right) \qquad A \sim \mathrm{Bernoulli}(\pi(V,Z))$$

where $\sigma(x) = \frac{1}{1+e^{-x}}$. We normalize $\pi(v,z)$ by $\frac{1}{\sqrt{k_v+k_z}}$ to satisfy Condition 2.1.4 and use the coefficient $k_v/(k_v + \rho k_z)$ to facilitate a fair comparison as we vary $\rho$. For all experiments, we report test MSE for 300 simulations where each simulation generates $n = 2000$ data points split randomly and evenly into train and test sets.[7] In the first set of experiments, for fixed $d = d_{\mathrm{V}} + d_{\mathrm{Z}} = 500$, we vary $d_{\mathrm{V}}$ (and correspondingly $d_{\mathrm{Z}}$). We also vary $k_z$, which governs the runtime confounding. Larger values of $k_z$ correspond to more confounding variables. The theoretical analysis (§ 3) suggests that when confounding ($k_z$) is small, then the TCR and DR methods will perform well. More confounding (larger $k_z$) should increase error for all methods, and we expect this increase to be significantly larger for the TCR method that has confounding bias. We expect the TCR and DR methods to perform better at smaller values of $d_{\mathrm{V}}$; by contrast, we expect the PL performance to vary less with $d_{\mathrm{V}}$ since the PL method suffers from the full $d$-dimensionality in the first stage regardless of $d_{\mathrm{V}}$. For large values of $d_{\mathrm{V}}$, we expect the PL method to perform similarly to the DR method. Fig. 1 plots the MSE in estimating $\nu$ for $\rho = 0$ and $k_v = 25$ using LASSO and random forests. The LASSO plots in Fig. 1a and 1b show the expected trends. Random forests have much higher error than the LASSO (compare Fig. 1a to 1c) and we only see a small increase in error as we increase confounding (Fig. 1c) because the random forest estimation error dominates the confounding error. In this setting, the TCR method may outperform the other methods, and in fact the TCR performs best at low levels of confounding.

We next consider the case were $V$ and $Z$ are correlated. If V and Z are perfectly correlated, there is no confounding. For our data where higher values of $V$ and $Z$ both decrease $\pi$ and increase $\mu$, a positive correlation should reduce confounding, and a negative correlation may exacerbate confounding by increasing the probability that $Z$ is small given $A = a$ and $V$ is large and therefore increasing the gap $\mathbb{E}[Y^a \mid V = v] - \mathbb{E}[Y^a \mid V = v, A = a]$. Fig. 2 gives MSE for correlated V and Z. As expected, error overall decreases with $\rho$ (Fig. 2a). Relative to the uncorrelated setting (Fig. 1), the weak positive correlation reduces MSE for all methods, particularly for large $k_z$ and $d_{\mathrm{V}}$. The DR method achieves the lowest error for settings with confounding, performing on par with the TCR when $d_{\mathrm{V}} = 50$.

**Experiments with Second-Stage Misspecification**  Next, we explore a more complex data generating process through the lens of model interpretability. Interpretability requirements allow for a complex training process as long as the final model outputs interpretable predictions [41, 49, 34]. Since the PL and DR first stage regressions are only a part of the training process, we can use any flexible model to learn the first stage functions as accurately as possible without impacting interpretability. Constraining the second-stage learning class to interpretable models (e.g. linear classifiers) may cause misspecification since the interpretable class may not contain the true model.

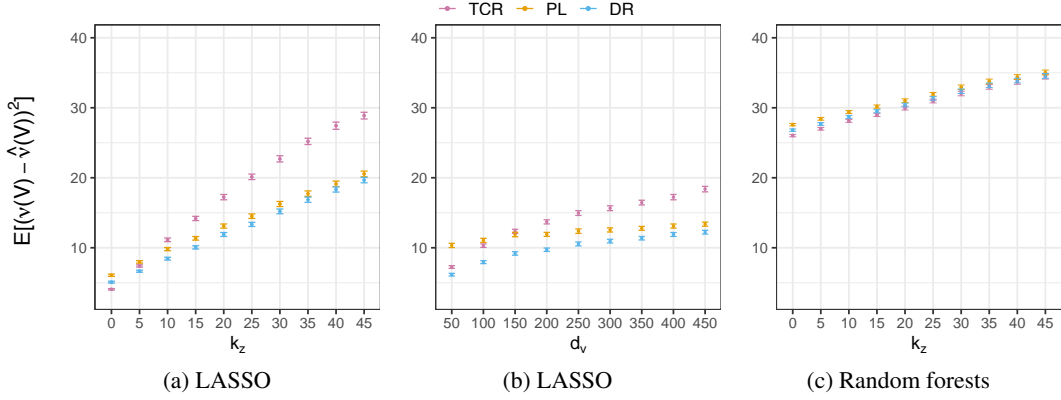

(a) LASSO         (b) LASSO         (c) Random forests

Figure 1: **(a)** MSE as we vary $k_z$ using cross-validated LASSO to learn $\hat{\pi}$, $\hat{\mu}$, $\hat{\nu}_{\text{TCR}}$, $\hat{\nu}_{\text{PL}}$, $\hat{\nu}_{\text{DR}}$ for $\rho = 0$, $d_{\text{V}} = 400$ and $k_v = 25$. At low levels of confounding ($k_z$), the TCR method does well but performance degrades with $k_z$. For any non-zero confounding, our DR method performs best.
**(b)** MSE against $d_{\text{V}}$ using cross-validated LASSO and $\rho = 0$, $k_v = 25$ and $k_z = 20$. The DR method performs the best across the range of $d_{\text{V}}$. When $d_{\text{V}}$ is small, the TCR method also does well since its estimation error is small. The PL method has higher error since it suffers from the full $d$-dimensional estimating error in the first stage. **(c)** MSE as we vary $k_z$ using random forests and $\rho = 0$, $d_{\text{V}} = 400$ and $k_v = 25$. Compared to LASSO in (a), there is a relatively small increase in error as we increase $k_z$, suggesting that estimation error dominates the confounding error. The TCR method performs best at lower levels of confounding and on par with the DR method for larger values of $k_z$.
Error bars denote 95% confidence intervals.

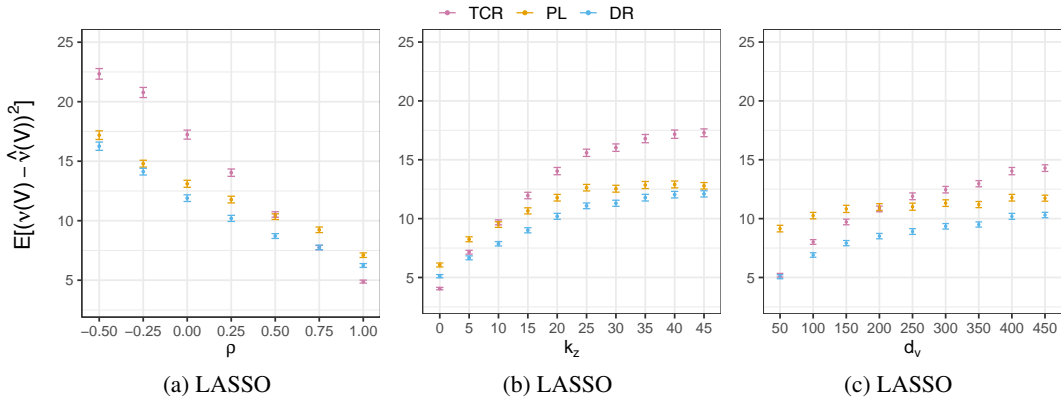

(a) LASSO         (b) LASSO         (c) LASSO

Figure 2: **(a)** MSE against correlation $\rho_{V_i, Z_i}$ for $k_z = 20$, $k_v = 25$, and $d_{\text{V}} = 400$. Error decreases with $\rho$ for all methods. Our DR method achieves the lowest error under confounding ($\rho < 1$). **(b)** MSE as we increase $k_z$ for $\rho = 0.25$, $k_v = 25$, and $d_{\text{V}} = 400$. Compare to Figure 1a; the weak positive correlation reduces MSE, particularly for $k_v < i \leq k_z$ when $V_i$ is only a correlate for the confounder $Z_i$ but not a confounder itself. **(c)** MSE against $d_{\text{V}}$ for $\rho = 0.25$, $k_z = 20$, and $k_v = 25$. The DR method is among the best-performing for all $d_{\text{V}}$. As with the uncorrelated setting (1b), the DR and TCR methods are better able to take advantage of low $d_{\text{V}}$ than the PL method.
Error bars denote 95% confidence intervals.

We simulate such a setting by modifying the setup (for $\rho = 0$):

$$V_i \sim \mathcal{N}(0,1) \ \text{ for } \ 1 \leq i \leq \frac{d_V}{2} \quad ; \quad V_i := V_j^2 \ \text{ for } \ \frac{d_V}{2} < i \leq d_V, \ \ j = i - \frac{d_V}{2}$$

$$\mu(V,Z) = \sum_{i=1}^{k_v/2} \left( V_i + (2(i \bmod 2) - 1)V_i^2 \right) + \sum_{i=1}^{k_z} Z_i \quad ; \quad \nu(V) = \sum_{i=1}^{k_v/2} \left( V_i + (2(i \bmod 2) - 1)V_i^2 \right)$$

We restrict our second stage models and the TCR model to predictors $V_i$ for $1 \leq i \leq \frac{d_V}{2}$ to simulate a real-world setting where we are constrained to linear classifiers using only $V$ at runtime. We allow the first stage models access to the full $V$ and $Z$ since the first stage is not constrained by variables or model class. We use cross-validated LASSO models for both stages and compare this setup to the setting where the model is correctly specified. The DR method achieves the lowest error for both settings (Table 1), although the error is significantly higher for all methods under misspecification.

| Method | Correct specification | 2nd-stage misspecification |
|---|---|---|
| TCR | 16.64 (16.28, 17.00) | 35.52 (35.18, 35.85) |
| PL | 12.32 (12.03, 12.61) | 32.09 (31.82, 32.36) |
| DR (ours) | **11.10 (10.84, 11.37)** | **31.33 (31.06, 31.59)** |

Table 1: MSE $\mathbb{E}\big[\big(\nu(V) - \hat{\nu}(V)\big)^2\big]$ under correct specification vs misspecification in the 2nd stage for $d = 500$, $d_V = 400$, $k_v = 24$, $k_z = 20$ and $n = 3000$ (with 95% confidence intervals). Our DR method has the lowest error in both settings. Errors are larger for all methods under misspecification.

### 5.1 Experiments on real-world child welfare data

In the US, each year over 4 million calls are made to child welfare screening hotlines with allegations of child neglect or abuse [42]. Call workers must decide which allegations coming in to the child abuse hotline should be investigated. In agencies that have adopted risk assessment tools, the worker relies on (immediate risk) information communicated during the call and an algorithmic risk score that summarizes (longer term) risk based on historical administrative data [9]. The call is recorded but is not used as a predictor for three reasons: (1) the inadequacy of existing case management software to run speech/NLP models on calls in realtime; (2) model interpretability requirements; and (3) the need to maintain distinction between immediate risk (as may be conveyed during the call) and longer-term risk the model seeks to estimate. Since it is not possible to use call information as a predictor, we encounter runtime confounding. Additionally, we would like to account for the disproportionate involvement of families of color in the child welfare system [12], but due to its sensitivity, we do not want to use race in the prediction model.

The task is to predict which cases are likely to be offered services under the decision $a =$ "screened in for investigation" using historical administrative data as predictors ($V$) and accounting for confounders race and allegations in the call ($Z$). Our dataset consists of over 30,000 calls to the hotline in Allegheny County, PA. We use random forests in the first stage for flexibility and LASSO in the second stage for interpretability. Table 2 presents the MSE using our evaluation method (§ 4).[8] The PL and DR methods achieve a statistically significant lower MSE than the TCR approach, suggesting these approaches could help workers better identify at-risk children than standard practice.

| | |
|---|---|
| TCR | 0.290 (0.287, 0.293) |
| PL | **0.249 (0.246, 0.251)** |
| DR (ours) | **0.248 (0.245, 0.250)** |

Table 2: MSE estimated via our evaluation procedure (§ 4) for child welfare screening task. The PL and DR approaches achieve lower MSE than the TCR approach. 95% confidence intervals given.

## 6 Conclusion

We propose a generic procedure for learning counterfactual predictions under runtime confounding that can be used with any parametric or nonparametric learning algorithm. Our theoretical and empirical analysis suggests this procedure will often outperform other methods, particularly when the level of runtime confounding is significant.

## Broader Impact

Real-world adoption of our proposed methodology may have a number of ethical and societal consequences. Our method is well-suited to decision support settings, including high-stakes decisions such as public assistance, parole and bail decisions in criminal justice, and treatment prioritization in healthcare. Our proposed method has the potential to improve decision-making in settings with runtime confounding where, as demonstrated in this paper, standard methods produce biased results. Beyond the statistical bias of simply failing to target the right counterfactual quantity, if decisions are made based on such predictions, it may disadvantage certain demographic groups in cases such as where group membership is a confounding factor in observed decisions [10]. This is a significant concern because group membership is often an impermissible input to decision-support tools at runtime, while also being a factor that influences discriminatory decision-making in observed data. Using our methods in these settings can improve predictions and the decisions they ultimately inform.

However, our proposed approach is valid only in the setting where our assumed Conditions 2.1 hold, and is not offered as a panacea for generally confounded data. The assumption that training data is unconfounded (§ 2.1.1) deserves considerable scrutiny any time the methods are applied. This assumption cannot be verified empirically and must instead be evaluated by domain experts who have detailed knowledge of the historical decision-making process. We encourage practitioners to carefully consider the validity of this assumption for their setting. Further data collection may be required to ensure that the data available for training does contain all factors that may have been relevant to historical decision-making, even if it is not information that is desirable or permissible to be used at runtime.

To illustrate the potential benefits as well as possible misuses of our method, we consider how our method could inform parole decisions. Parole boards determine whether and under what conditions to release a person from incarceration. Recidivism risk assessment models are widely adopted by probation and parole departments around the US. It is often of interest to assess the likelihood of success under different possible supervision conditions. Runtime confounding occurs in the setting when, for instance, the parole board makes a recommendation after reviewing documents and hearing spoken testimony, but the board would like to see the predictions of a risk and needs assessment tool prior to the hearing. The testimony may provide information that both influences the board's decision and reveals drivers of the offender's likelihood to succeed if released, but this information is unavailable at prediction time, leading to runtime confounding. Moreover, we may be concerned that parole boards implicitly used race to make decisions and would like to account for this without requiring the use of race as a model input. Our method would allow us to do so. Our method can handle some of the challenging aspects of this setting, but there may be other problems that are not addressed by our method. For instance, while our method can help account for racial bias in historical parole decisions, it cannot correct for racial bias in the downstream outcomes. Since research suggests that people of color are disproportionately arrested relative to true crime rates [1], one should be wary of using these outcomes. Predictive models trained on such outcomes could perpetuate or exacerbate racial disparities in criminal justice. Additionally, if Conditions 2.1 do not hold because e.g. the spoken testimony is not accurately recorded, then our method may lead to unreliable predictions.

The appropriate use of our method in high-stakes real-world settings would include careful consideration of the validity of Conditions 2.1 as well as other potential biases in the data used for model training. If deployed in the appropriate setting, our method has the potential to help decision-makers make better decisions that can improve efficiency and fairness.

## Acknowledgments and Disclosure of Funding

This research was made possible through support from Tata Consultancy Services (TCS) Presidential Fellowship, the K&L Gates Presidential Fellowship, and the Block Center for Technology and Society. This material is based upon work supported by the National Science Foundation Grants No. DMS1810979, IIS1939606, and the National Science Foundation Graduate Research Fellowship Program under Grant No. DGE1745016. Any opinions, findings, and conclusions or recommendations expressed in this material are those of the author(s) and do not necessarily reflect the views of the National Science Foundation. We are grateful to Allegheny County Department of Human Services for sharing their data. Thanks to our reviewers for providing useful feedback about the project and to Siddharth Ancha for helpful discussions.

## Footnotes

[1] For exposition, we focus on making predictions for a single binary treatment $a$. To make predictions under multiple discrete treatments, our method can be repeated for each treatment using a one-vs-all setup.

[2] Runtime imputation of $Z$ will not eliminate this bias since $E[Y \mid A = a, V = v, f(v)] = \omega(v)$.

[3] The term *nuisance* refers to functions $\mu$ and $\pi$.

[4] We use the sparsity parameter $k$ to indicate $k$ covariates have non-zero coefficients in the model.

[5]The appendix describes a cross-fitting approach to jointly learn and evaluate the three prediction methods.

[6]We report error metrics on a random heldout test set.

[7]Source code is in the appendix and will be available at *https://github.com/*

[8]We report error metrics on a random heldout test set.

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
