[Supplementary Material]

# Counterfactual Predictions under Runtime Confounding
# Supplementary Material

**Amanda Coston**
Heinz College & Machine Learning Department
Carnegie Mellon University
acoston@cs.cmu.edu

**Edward H. Kennedy**
Department of Statistics
Carnegie Mellon University
edward@stat.cmu.edu

**Alexandra Chouldechova**
Heinz College
Carnegie Mellon University
achould@cmu.edu

## A  Details on Proposed Learning Procedure

We describe a joint approach to learning and evaluating the TCR, PL, and DR prediction methods in Algorithm 6. This approach efficiently makes use of the need for both prediction and evaluation methods to estimate the propensity score $\pi$.

---

**Algorithm 6** Cross-fitting procedure to learn and evaluate the TCR, PL, and DR prediction methods

---

**Input:** Data samples $\{(V_j, Z_j, A_j, Y_j)\}_{j=1}^{4n}$

Randomly divide training data into four partitions $\mathcal{W}^1, \mathcal{W}^2, \mathcal{W}^3, \mathcal{W}^4$ where $\mathcal{W}^1 = \{(V_j^1, Z_j^1, A_j^1, Y_j^1)\}_{j=1}^n$ (and similarly for $\mathcal{W}^2$, $\mathcal{W}^3, \mathcal{W}^4$).

**for** $(p, q, r, s) \in \{(1,2,3,4), (4,1,2,3), (3,4,1,2), (2,3,4,1)\}$ **do**

    *Stage 1:* On $\mathcal{W}^p$, learn $\hat{\mu}^p(v, z)$ by regressing $Y \sim V, Z \mid A = a$.

        On $\mathcal{W}^q$, learn $\hat{\pi}^q(v, z)$ by regressing $\mathbb{I}\{A = a\} \sim V, Z$

    *Stage 2:* On $\mathcal{W}^r$, learn $\hat{\nu}_{\mathrm{DR}}^r$ by regressing $\left( \frac{\mathbb{I}\{A=a\}}{\hat{\pi}^q(V,Z)} (Y - \hat{\mu}^p(V,Z)) + \hat{\mu}^p(V,Z) \right) \sim V$

        On $\mathcal{W}^r$ and $\mathcal{W}^q$, learn $\hat{\nu}_{\mathrm{PL}}^r$ by regressing $\hat{\mu}^p(V,Z) \sim V$

        On $\mathcal{W}^r$, $\mathcal{W}^q$, and $\mathcal{W}^p$, learn $\hat{\nu}_{\mathrm{TCR}}^r$ by regressing $Y \sim V \mid A = a$

    *Evaluate* for $m$ in { TCR, PL, DR }:

        On $\mathcal{W}^q$, learn $\hat{\eta}_m^q(v, z)$ by regressing $(Y - \hat{\nu}_m^r(V))^2 \sim V, Z \mid A = a$

        On $\mathcal{W}^s$, for $j = 1, \dots n$ compute $\phi_{m,j}^s = \frac{\mathbb{I}\{A_j = a\}}{\hat{\pi}^q(V_j, Z_j)} ((Y_j - \hat{\nu}_m^r(V_j))^2 - \hat{\eta}_m^q(V_j, Z_j)) + \hat{\eta}_m^q(V_j, Z_j)$

**Output prediction models:** $\hat{\nu}_{\mathrm{DR}}(v) = \frac{1}{4} \sum_{j=1}^4 \hat{\nu}_{\mathrm{DR},j}(v))$;    $\hat{\nu}_{\mathrm{PL}}(v) = \frac{1}{4} \sum_{j=1}^4 \hat{\nu}_{\mathrm{PL},j}(v)$ ;    $\hat{\nu}_{\mathrm{TCR}}(v) = \frac{1}{4} \sum_{j=1}^4 \hat{\nu}_{\mathrm{TCR},j}(v)$

**Output error estimate confidence intervals:** for $m$ in { TCR, PL, DR }:

$$\mathrm{MSE}_m = \left( \frac{1}{4n} \sum_{i=1}^4 \sum_{j=1}^n \phi_{m,j}^i \right) \pm 1.96 \sqrt{\frac{1}{4n} \mathrm{var}(\phi_m)}$$

---

## B  Proofs and derivations

In this section we provided detailed proofs and derivations for all results in the main paper.

## B.1 Derivation of Identifications of $\mu$ and $\nu$

We first show the steps to identify $\mu(v, z)$:

$$\mu(v, z) = \mathbb{E}[Y^a \mid V = v, Z = z]$$
$$\mathbb{E}[Y^a \mid V = v, Z = z] = \mathbb{E}[Y^a \mid V = v, Z = z, A = a]$$
$$= \mathbb{E}[Y \mid V = v, Z = z, A = a]$$

The first line applies the definition of $\mu$.. The second line follows from training ignorability (Condition 2.1.1). The third line follows from consistency (Condition 2.1.3).

Next we show the identification of $\nu(v)$:

$$\nu(v) = \mathbb{E}[Y^a \mid V = v]$$
$$\mathbb{E}[Y^a \mid V = v] = \mathbb{E}[\mathbb{E}[Y^a \mid V = v, Z = z] \mid V = v]$$
$$= \mathbb{E}[\mathbb{E}[Y^a \mid V = v, Z = z, A = a] \mid V = v]$$
$$= \mathbb{E}[\mathbb{E}[Y \mid V = v, Z = z, A = a] \mid V = v]$$

The first line applies the definition of $\nu$ from Section 2. The second line follows from iterated expectation. The third line follows from training ignorability (Condition 2.1.1). The fourth line follows from consistency (Condition 2.1.3).

Note that we can concisely rewrite the last line as $\mathbb{E}[\mu(V, Z) \mid V = v]$ since we have identified $\mu$.

## B.2 Proof that TCR method underestimates risk under mild assumptions on a risk assessment setting

*Proof.* In Section 3.1 we posited that the TCR method will often underestimate risk in a risk assessment setting. We demonstrate this for the setting with a binary outcome $Y \in \{0, 1\}$, but the logic extends to settings with a discrete or continuous outcome. We assume larger values of $Y$ are adverse i.e. $Y = 0$ is desired and $Y = 1$ is adverse. The decision under which we'd like to estimate outcomes is the baseline decision $A = 0$. We start by recalling runtime confounding condition (2.1.2): $\mathbb{P}(A = 0 \mid V, Y^0 = 1) \neq \mathbb{P}(A = 0 \mid V, Y^0 = 0)$. Here we further refine this by assuming we are in the common setting where treatment $A = 1$ is more likely to be assigned to people who are higher risk. Then $\mathbb{P}(A = 1 \mid V, Y^0 = 1) > \mathbb{P}(A = 1 \mid V, Y^0 = 0)$. Equivalently $\mathbb{P}(A = 0 \mid V, Y^0 = 1) < \mathbb{P}(A = 0 \mid V, Y^0 = 0)$. By the law of total probability,

$$\mathbb{P}(A = 0 \mid V) = \mathbb{P}(A = 0 \mid V, Y^0 = 1)\mathbb{P}(Y^0 = 1 \mid V) + \mathbb{P}(A = 0 \mid V, Y^0 = 0)\mathbb{P}(Y^0 = 0 \mid V)$$

Assuming $\mathbb{P}(Y^0 = 1 \mid V) > 0$, this implies

$$\mathbb{P}(A = 0 \mid V, Y^0 = 0) > \mathbb{P}(A = 0 \mid V) \tag{2}$$

By Bayes' rule,

$$\mathbb{P}(A = 0 \mid V, Y^0 = 0) = \mathbb{P}(Y^0 = 0 \mid V, A = 0)\frac{\mathbb{P}(A = 0 \mid V)}{\mathbb{P}(Y^0 = 0 \mid V)}$$

Using this in the LHS of Equation 2 and dividing both sides of Equation 2 by $\mathbb{P}(A = 0 \mid V)$, we get

$$\frac{\mathbb{P}(Y^0 = 0 \mid V, A = 0)}{\mathbb{P}(Y^0 = 0 \mid V)} > 1$$

Equivalently $\mathbb{E}[Y^0 \mid V, A = 0] < \mathbb{E}[Y^0 \mid V]$. $\qquad\square$

## B.3 Derivation of Proposition 3.1 (confounding bias of the TCR method)

We recall **Proposition 3.1**:
Under runtime confounding, a model that perfectly predicts $\omega(v)$ has pointwise confounding bias $b(v) = \omega(v) - \nu(v) =$

$$\int_{\mathcal{Z}} \mu(v, z)\Big(p(z \mid V = v, A = a) - p(z \mid V = v)\Big)dz \quad \neq \quad 0 \tag{3}$$

*Proof.* By iterated expectation and the definition of expectation we have that

$$\omega(v) = \int_{\mathcal{Z}} \mathbb{E}[Y \mid V = v, Z = z, A = a]\, p(z \mid V = v, A = a) dz$$

$$= \int_{\mathcal{Z}} \mu(v, z) p(z \mid V = v, A = a) dz$$

In the identification derivation above we saw that $\nu(v) = \mathbb{E}[\mu(V, Z) \mid V = v]$. Using the definition of expectation, we can rewrite $\nu(v)$ as

$$= \int_{\mathcal{Z}} \mu(v, z) p(z \mid V = v) dz$$

Therefore the pointwise bias is

$$\omega(v) - \nu(v) = \int_{\mathcal{Z}} \mu(v, z) \Big( p(z \mid V = v, A = a) - p(z \mid V = v) \Big) dz \qquad (4)$$

We can prove that this pointwise bias is non-zero by contradiction. Assuming the pointwise bias is zero, we have $\omega(v) = \nu(v) \implies Y^a \perp A \mid V = v$ which contradicts the runtime confounding condition 2.1.2. $\square$

We emphasize that the confounding bias does not depend on the treatment effect. This approach is problematic whenever treatment assignment depends on $Y^a$ to an extent that is not measured by $V$, even for settings with no treatment effect (such as selective labels setting [21, 20]).

## B.4 Proof of Proposition 3.2 (error of the TCR method)

We can decompose the pointwise error of the TCR method into the estimation error and the bias of the TCR target.

*Proof.*

$$\mathbb{E}[(\nu(v) - \hat{\nu}_{\text{TCR}}(v))^2] = \mathbb{E}\Big[\Big((\nu(v) - \omega(v)) + (\omega(v) - \hat{\nu}_{\text{TCR}}(v))\Big)^2\Big]$$

$$\leq 2\Bigg(\mathbb{E}[(\nu(v) - \omega(v))^2] + \mathbb{E}[(\omega(v) - \hat{\nu}_{\text{TCR}}(v))^2]\Bigg)$$

$$\lesssim (\nu(v) - \omega(v))^2 + \mathbb{E}[(\omega(v) - \hat{\nu}_{\text{TCR}}(v))^2]$$

$$= b(v)^2 + \mathbb{E}[(\omega(v) - \hat{\nu}_{\text{TCR}}(v))^2]$$

Where the second line is due to the fact that $(a + b)^2 \leq 2(a^2 + b^2)$. In the third line, we drop the expectation on the first term since there is no randomness in two fixed functions of $v$. $\square$

## B.5 Proofs of Proposition 3.3 and Theorem 3.1 (error of the PL and DR methods)

We begin with additional notation needed for the proofs of the error bounds. For brevity let $W = (V, Z, A, Y)$ indicate a training observation. The theoretical guarantees for our methods rely on a two-stage training procedure that assumes independent training samples. We denote the first-stage training dataset as $\mathcal{W}^1 := \{W_1^1, W_2^1, W_3^1, \dots W_n^1\}$ and the second-stage training dataset as $\mathcal{W}^2 := \{W_1^2, W_2^2, W_3^2, \dots W_n^2\}$. Let $\hat{\mathbb{E}}_n[Y \mid V = v]$ denote an estimator of the regression function $\mathbb{E}[Y \mid V = v]$. Let $L \asymp R$ denote $L \lesssim R$ and $R \lesssim L$.

**Definition B.1.** (Stability conditions) The results assume the following two stability conditions from [17] on the second-stage regression estimators:

**Condition B.1.1.** $\hat{\mathbb{E}}_n[Y \mid V = v] + c = \hat{\mathbb{E}}_n[Y + c \mid V = v]$ for any constant $c$

**Condition B.1.2.** For two random variables $R$ and $Q$, if $\mathbb{E}[R \mid V = v] = \mathbb{E}[Q \mid V = v]$, then

$$\mathbb{E}\left[\left(\hat{\mathbb{E}}_n[R \mid V = v] - \mathbb{E}[R \mid V = v]\right)^2\right] \asymp \mathbb{E}\left[\left(\hat{\mathbb{E}}_n[Q \mid V = v] - \mathbb{E}[Q \mid V = v]\right)^2\right]$$

### B.5.1 Proof of Proposition 3.3 (error of thePL method)

The theoretical results for our two-stage procedures rely on the theory for pseudo-outcome regression in Kennedy [17] which bounds the error for a two-stage regression on the full set of confounding variables. However, our setting is different since our second-stage regression is on a subset of confounding variables. Therefore, Theorem 1 of Kennedy [17] does not immediately give the error bound for our setting, but we can use similar techniques in order to get the bound for our V-conditional second-stage estimators.

*Proof.* As our first step, we define an error function. The error function of the PL approach is $\hat{r}_{\mathrm{PL}}(v)$

$$
\begin{aligned}
&= \mathbb{E}[\hat{\mu}(V, Z) \mid V = v, \mathcal{W}^1] - \nu(v) \\
&= \mathbb{E}[\hat{\mu}(V, Z) \mid V = v, \mathcal{W}^1] - \mathbb{E}[\mu(V, Z) \mid V = v] \\
&= \mathbb{E}[\hat{\mu}(V, Z) - \mu(V, Z) \mid V = v, \mathcal{W}^1]
\end{aligned}
$$

The first line is our definition of the error function (following [17]). The second line uses iterated expectation, and the third lines uses the fact that $\mathcal{W}^1$ is a random sample of the training data. Next we square the error function and apply Jensen's inequality to get

$$
\hat{r}_{\mathrm{PL}}(v)^2 = \left( \mathbb{E}[\hat{\mu}(V, Z) - \mu(V, Z) \mid V = v, \mathcal{W}^1] \right)^2 \leq \mathbb{E}\left[ \left( \hat{\mu}(V, Z) - \mu(V, Z) \right)^2 \mid V = v, \mathcal{W}^1 \right]
$$

Taking the expectation over $\mathcal{W}^1$ on both sides, we get

$$
\begin{aligned}
\mathbb{E}[\hat{r}_{\mathrm{PL}}(v)^2 \mid V = v] &\leq \mathbb{E}\left[ \mathbb{E}\left[ \left( \hat{\mu}(V, Z) - \mu(V, Z) \right)^2 \mid V = v, \mathcal{W}^1 \right] \mid V = v \right] \\
&= \mathbb{E}\left[ \left( \hat{\mu}(V, Z) - \mu(V, Z) \right)^2 \mid V = v \right]
\end{aligned}
$$

Next, under our stability conditions(§ B.1), we can apply Theorem 1 of Kennedy [17] (stated in the next section for reference) to get the pointwise bound

$$
\mathbb{E}\left[ \left( \hat{\nu}_{\mathrm{PL}}(v) - \nu(v) \right)^2 \right] \lesssim \mathbb{E}\left[ \left( \tilde{\nu}(v) - \nu(v) \right)^2 \right] + \mathbb{E}\left[ \left( \hat{\mu}(V, Z) - \mu(V, Z) \right)^2 \mid V = v \right]
$$

Theorem 1 of Kennedy also implies a bound on the integrated MSE of the PL approach:

$$
\mathbb{E}\left\| \hat{\nu}_{\mathrm{PL}}(v) - \nu(v) \right\|^2 \lesssim \mathbb{E}\left\| \tilde{\nu}(v) - \nu(v) \right\|^2 + \int_{\mathcal{V}} \mathbb{E}\left[ (\hat{\mu}(V, Z) - \mu(V, Z))^2 \mid V = v \right] p(v) dv
$$

$\square$

### B.5.2 Theorem for Pseudo-Outcome Regression (Kennedy)

The proofs of Proposition 3.3 and Theorem 3.1 rely on Theorem 1 of Kennedy [17] which we restate here for reference. In what follows we provide the proof for Theorem 3.1.

**Theorem B.1** (Kennedy). Recall that $\mathcal{W}^1$ denotes our $n$ first-stage training data samples. Let $\hat{f}(w) := \hat{f}(w; \mathcal{W}^1)$ be an estimate of the function $f(w)$ using the training data $\mathcal{W}^1$. Denote an independent sample as $W$. The true regression function is $m(v) := \mathbb{E}[f(W) \mid V = v]$. Denote the second stage regression as $\hat{m}(v) := \hat{\mathbb{E}}_n[\hat{f}(W) \mid V = v]$. Denote its oracle equivalent (if we had access to $Y^a$) as $\tilde{m}(v) := \hat{\mathbb{E}}_n[f(W) \mid V = v]$. Under stability conditions(§ B.1) on the regression estimator $\hat{\mathbb{E}}_n$, we have the following bound on the pointwise MSE:

$$
\mathbb{E}\left[ \left( \hat{m}(v) - m(v) \right)^2 \right] \lesssim \mathbb{E}\left[ \left( \tilde{m}(v) - m(v) \right)^2 \right] + \mathbb{E}\left[ \hat{r}(v)^2 \right]
$$

where $\hat{r}(v)$ describes the error function $\hat{r}(v) := \mathbb{E}[\hat{f}(W) \mid V = v, \mathcal{W}^1] - m(v)$. This implies the following bound for the integrated MSE:

$$
\mathbb{E}\left\| \hat{m}(v) - m(v) \right\|^2 \lesssim \mathbb{E}\left\| \tilde{m}(v) - m(v) \right\|^2 + \int \mathbb{E}[\hat{r}(v)^2] p(v) dv
$$

### B.5.3 Proof of Theorem 3.1 (error of the DR method)

Here we provide the proof for our main theoretical result which bounds the error of our proposed DR method.

*Proof.* As for the PL error bound above, the first step is to derive the form of the error function for our DR approach. For clarity and brevity, we denote the measure of the expectation in the subscript.

$$\hat{r}_{\text{DR}}(v) = \mathbb{E}_{W|V=v,\mathcal{W}^1}\left[\frac{\mathbb{I}\{A=a\}}{\hat{\pi}(v,Z)}(Y-\hat{\mu}(v,Z)) + \hat{\mu}(v,Z)\right] - \nu(v)$$

$$= \mathbb{E}_{Z,A|V=v,\mathcal{W}^1}\left[\mathbb{E}_{W|A=a,V=v,Z=z,\mathcal{W}^1}\left[\frac{\mathbb{I}\{A=a\}}{\hat{\pi}(v,Z)}(Y-\hat{\mu}(v,z)) + \hat{\mu}(v,z)\right]\right] - \nu(v)$$

$$= \mathbb{E}_{Z,A|V=v,\mathcal{W}^1}\left[\mathbb{E}_{Y|A=a,V=v,Z=z,\mathcal{W}^1}\left[\frac{\mathbb{I}\{A=a\}}{\hat{\pi}(v,Z)}(Y-\hat{\mu}(v,z))\right] + \hat{\mu}(v,Z)\right] - \nu(v)$$

$$= \mathbb{E}_{Z,A|V=v,\mathcal{W}^1}\left[\frac{\mathbb{I}\{A=a\}}{\hat{\pi}(v,Z)}(\mathbb{E}_{Y|A=a,V=v,Z=z,\mathcal{W}^1}[Y]-\hat{\mu}(v,Z)) + \hat{\mu}(v,Z)\right] - \nu(v)$$

$$= \mathbb{E}_{W|V=v,\mathcal{W}^1}\left[\frac{\mathbb{I}\{A=a\}}{\hat{\pi}(v,Z)}(\mu(v,Z)-\hat{\mu}(v,Z)) + \hat{\mu}(v,Z)\right] - \nu(v)$$

$$= \mathbb{E}_{Z|V=v,,\mathcal{W}^1}\left[\mathbb{E}_{W|V=v,Z=z,\mathcal{W}^1}\left[\frac{\mathbb{I}\{A=a\}}{\hat{\pi}(v,Z)}(\mu(v,z)-\hat{\mu}(v,z)) + \hat{\mu}(v,z)\right]\right] - \nu(v)$$

$$= \mathbb{E}_{Z|V=v,\mathcal{W}^1}\left[\frac{\mathbb{P}(A=a \mid V=v,Z=z)}{\hat{\pi}(v,Z)}(\mu(v,Z)-\hat{\mu}(v,Z)) + \hat{\mu}(v,Z)\right] - \nu(v)$$

$$= \mathbb{E}_{Z|V=v,\mathcal{W}^1}\left[\frac{\pi(v,Z)}{\hat{\pi}(v,Z)}(\mu(v,Z)-\hat{\mu}(v,Z)) + \hat{\mu}(v,Z)\right] - \nu(v)$$

$$= \mathbb{E}_{Z|V=v,\mathcal{W}^1}\left[\frac{\pi(v,Z)}{\hat{\pi}(v,Z)}(\mu(v,Z)-\hat{\mu}(v,Z)) + \hat{\mu}(v,Z) - \mu(v,Z)\right]$$

$$= \mathbb{E}\left[\frac{(\mu(v,Z)-\hat{\mu}(v,Z))(\pi(v,Z)-\hat{\pi}(v,Z))}{\hat{\pi}(v,Z)} \mid V=v,\mathcal{W}^1\right]$$

Where the first line holds by definition of the error function $\hat{r}$ and the second line by iterated expectation. The third line uses the fact that conditional on $Z=z, V=v, A=a$, then the only randomness in $W$ is $Y$ (and therefore $\hat{\mu}$ is constant). The fourth line makes use of the $(\mathbb{I}\{A=a\})$ term to allow us to condition on only $A=a$ ( since the term conditioning on any other $a' \neq a$ will evaluate to zero). The fifth line applies the definition of $\mu$.

The sixth line again uses iterated expectation and the seventh makes use of the fact that conditional on $Z$, the only randomness now is in $A$ and that $\mathcal{W}^1$ is an independent randomly sampled set. The seventh line applies the definition of $\pi(v,z) = \mathbb{P}(A=1 \mid V=v,Z=z)$ which since $A \in \{0,1\}$ is equal to $\mathbb{E}[A \mid V=v,Z=z]$. The eight line uses iterated expectation and the fact that $\mathcal{W}^1$ is an independent randomly sampled set to rewrite $\nu(v) = E_{Z|V=v,\mathcal{W}^1}[\mu(v,Z)]$. The ninth line rearranges the terms.

By Cauchy-Schwarz and the positivity assumption,

$$\hat{r}_{\text{DR}}(v) \leq C\sqrt{\mathbb{E}[(\mu(v,Z)-\hat{\mu}(v,Z))^2 \mid V=v,\mathcal{W}^1]}\sqrt{\mathbb{E}[(\pi(v,Z)-\hat{\pi}(v,Z))^2 \mid V=v,\mathcal{W}^1]}$$

for a constant $C$.

Squaring both sides yields

$$\hat{r}_{\text{DR}}^2(v) \leq C^2 \, \mathbb{E}[(\mu(v,Z)-\hat{\mu}(v,Z))^2 \mid V=v,\mathcal{W}^1] \, \mathbb{E}[(\pi(v,Z)-\hat{\pi}(v,Z))^2 \mid V=v,\mathcal{W}^1]$$

If $\hat{\pi}$ and $\hat{\mu}$ are estimated using separate training samples, then taking the expectation over the first-stage training sample $\mathcal{W}^1$ yields:

$$\mathbb{E}[\hat{r}_{\mathrm{DR}}^2(v)] \leq C^2 \, \mathbb{E}[(\mu(v,Z) - \hat{\mu}(v,Z))^2] \mid V = v] \, \mathbb{E}[(\pi(v,Z) - \hat{\pi}(v,Z))^2] \mid V = v]$$

Applying Theorem 1 of Kennedy [17] gets the pointwise bound:

$$\mathbb{E}\left[\left(\hat{\nu}_{\mathrm{DR}}(v) - \nu(v)\right)^2\right] \lesssim \mathbb{E}\left[\left(\tilde{\nu}(v) - \nu(v)\right)^2\right]$$
$$+ \mathbb{E}\left[(\hat{\pi}(V,Z) - \pi(V,Z))^2 \mid V = v\right] \mathbb{E}\left[(\hat{\mu}(V,Z) - \mu(V,Z))^2 \mid V = v\right]$$

and the bound on integrated MSE of the DR approach:

$$\mathbb{E}\left\|\hat{\nu}_{\mathrm{DR}}(v) - \nu\right\|^2 \lesssim \mathbb{E}\left\|\tilde{\nu}(v) - \nu(v)\right\|^2$$
$$+ \int_{\mathcal{V}} \mathbb{E}\left[(\hat{\pi}(V,Z) - \pi(V,Z))^2 \mid V = v\right] \mathbb{E}\left[(\hat{\mu}(V,Z) - \mu(V,Z))^2 \mid V = v\right] p(v) dv$$

$\square$

## B.6 Efficient influence function for DR method

We provide the efficient influence function of the DR method. The efficient influence function indicates the form of the bias-correction term in the DR method. The efficient influence function $\phi(A, V, Z, Y)$ for parameter $\psi(V) := \mathbb{E}[Y^a \mid V] = \mathbb{E}[\mathbb{E}[Y \mid V, Z, A = a] \mid V]$ is

$$\phi(A, V, Z, Y) = \frac{\mathbb{I}\{A = a\}}{\pi(V,Z)}(Y - \mu(V,Z)) + \mu(V,Z) - \psi(V)$$

## C  Synthetic experiment details and additional results

In this section we present details on the synthetic experiments and present additional results. We present the random forests graphs omitted from the main paper, results on calibration-type curves that show where the errors are distributed, and experiments on our evaluation procedure.

### C.1  Experimental details

**More details on data-generating process**  We designed our data-generating process in order to simulate a real-world risk assessment setting. We consider both $V$ and $Z$ to be risk factors whose larger values indicate increased risk and therefore we construct $\mu$ to increase with $V$ and $Z$. Our goal is to assess risk under the null (or baseline) treatment as per [10], and we construct $\pi$ such that historically the riskier treatments were more likely to get the risk-mitigating treatment and the less risky cases were more likely to get the baseline treatment.

We now provide further details on the choices of coefficients and variance parameters. In the first set of experiments presented in the main paper, we simulate $V_i$ from a standard normal, and in the uncorrelated setting (where $\rho = 0$) we also simulate $Z_i$ from a standard normal. In the correlated setting, we sample $Z_i$ from a normal with mean $\rho V_i$ and variance $1 - \rho^2$ so that the Pearson's correlation coefficient between $V_i$ and $Z_i$ is $\rho$ and so that the variance in $Z_i = 1$. We simulate $\mu$ to be a sparse linear model in $V$ and $Z$ with coefficients of 1 when $\rho = 0$. When $\rho \neq 1$, the coefficients are set to $\frac{k_v}{k_v + \rho k_z}$ so that the $L_1$ norm of the $\nu$ coefficients equals $k_v$ for all values of $\rho$. Without this adjustment, changing $\rho$ would impact error by also changing the signal-to-noise ratio in $\nu$. We simulate the potential outcome $Y^a$ to be conditionally Gaussian and the choice of variance $\frac{1}{2n} \|\mu(V,Z)\|_2^2$ yields a signal-to-noise ratio of 2. The specification for $\nu$ follows from the marginalization of $\mu$ over $Z$. The propensity score $\pi$ depends on the sigmoid of a sparse linear function in $V$ and $Z$ that uses coefficients $\frac{1}{\sqrt{k_v + k_z}}$ in order to satisfy our positivity condition.

We use $d = 500$, $n = 1000$, $k_v = 25$, and $0 \leq k_z \leq 45$ to simulate a sparse high-dimensional setting with many measured variables in the training data, of which only 5%-15% are predictive of the outcomes. In one set of experiments, we vary the value of $k_z$ to assess impact of various levels of confounding on performance. In other experiments, where we vary $\rho$ or the dimensionality of $V$ ($d_V$), we use $k_z = 20$ so that $V$ has slightly more predictive power than the hidden confounders $Z$.

**Hyperparameters**   Our LASSO presents are presented for cross-validated hyperparameter selection using the `glmnet` package in R. The random forests results use 1000 trees and default *mtry* and splitting parameters in the `ranger` package in R.

**Training runs**   Defining a training run as performing a learning procedure such as LASSO, for a given hyperparameter selection and given simulation, the TCR method trains in one run, the PL method trains in two runs, and the DR method trains in three runs. For a given simulation, the exact number of runs depends on the hyperparameter tuning. Since we only ran random forests (RF) for the default parameters, the TCR method with RF trained in one run, the PL method with RF trained in two runs, and the DR method with RF trained in three runs. The LASSO results using `cv.glmnet` were tuned over $\leq 100$ values of $\lambda$; the TCR method with LASSO trained in $\leq 100$ runs, the PL method with LASSO trained in $\leq 200$ runs, and the DR method with LASSO trained in $\leq 300$ runs.

**Sample size and error metrics**   For experiments in the main paper, we trained on $n = 1000$ datapoints. We test on a separate set of $n = 1000$ datapoints and report the estimated mean squared error (MSE) on this test set using the following formula:

$$\frac{1}{n} \sum_{i=1}^{n} (\nu(V_i) - \hat{\nu}(V_i))^2$$

**Computing infrastructure**   We ran experiments on an Amazon Web Serivces (AWS) c5.12xlarge machine. This parallel computing environment was useful because we ran thousands of simulations. The traintime of each simulation, entailing the LASSO and RF experiments, took 1.8 seconds. In practice for most real-world decision support settings, our method can be used in standard computing environments; relative to existing predictive modeling techniques, our method will require $\leq 3X$ the current train time. Our runtime depends only on the regression technique used in the second stage and should be competitive to existing models.

## C.2   Random forest results

Figure 3 presents the results when using random forests for the first and second stage estimation in the uncorrelated V-Z setting. Figure 3a was provided in the main paper, and we include it here again for ease of reference. Figure 3b shows how method performance varies with $d_V$. At low $d_V$, the TCR method does significantly better than the two counterfactually valid approaches. This suggests that the estimation error incurred by the PL and DR methods outweighs the confounding bias of the TCR method.

## C.3   Evaluation experiments

To empirically assess our proposed doubly-robust evaluation procedure, we generated one sample of training data with $n = 1000$, $d = 500$, $d_V = 200$, $k_v = 25$, and $k_z = 30$ as well as a "ground-truth" test set with $n = 10,000$. We trained the TCR, PL, and DR methods on the training data and estimated their true performance on the large test set. The true prediction error

$$\frac{1}{n} \sum_{i=1}^{n} \left( Y_i^a - \hat{\nu}(V_i) \right)^2$$

was 77.53, 74.12, and 72.68 respectively for the TCR, PL and DR methods. We then ran 100 simulations where we sampled a more realistically sized test set of $n = 2000$. In each simulation we performance the evaluation procedure to estimate prediction error on the observed data. The MSE estimator with $95\%$ CI covered the true MSE 94 times for the DR approach and 93 times for the PL. 81% of the simulations correctly identified the DR procedure as having the lowest error, $14\%$ suggested that the PL procedure had the lowest error and $5\%$ suggested that the TCR had the lowest error.

For additional experimental results on using doubly-robust evaluation methods for predictive models, we recommend [10].

Figure 3: **(a)** ... DR for $\rho = 0$, $d_V = 400$ a... **(b)** MSE ag... Error bars d...

(a) Random forests     (b) Random forests     (c) Random forests

Figure 4: **(a)** MSE against correlation $\rho_{V_i, Z_i}$ for $k_z = 20$, $k_v = 25$, and $d_V = 400$. For all methods, error decreases with $\rho \leq 0.5$, at which point the error does not change with increasing $\rho$. **(b)** MSE as we increase $k_z$ for $\rho = 0.25$, $k_v = 25$, and $d_V = 400$. Compare to Figure 3a; the weak positive correlation reduces MSE, particularly for $k_v < i \leq k_z$ when $V_i$ is only a correlate for the confounder $Z_i$ but not a confounder itself. **(c)** MSE against $d_V$ for $\rho = 0.25$, $k_z = 20$, and $k_v = 25$. As with the uncorrelated setting (3b), the DR and TCR methods are better able to take advantage of low $d_V$ than the PL method.

Error bars denote $95\%$ confidence intervals.

Figure 5: (a) Calibration plot for LASSO regressions with $p = 400$, $q = 100$, $k_z = 20$ and $k_v = 25$. A well-calibrated model will track the dotted $y = x$ line. Our DR model is the best calibrated. As expected from its confounding bias, the TCR method underestimates risk for all predicted values. Interestingly the PL and DR methods also underestimate risk for higher predicted risk values.

(b) Squared error against true risk $\nu(V)$ for LASSO regressions with $p = 400$, $q = 100$, $k_z = 20$ and $k_v = 25$. All models have highest error on the riskiest cases (those with large values of $\nu(V)$); this is particularly pronounced for the TCR model, suggesting that the TCR model would make misleading predictions for the highest risk cases.

## C.4  Calibration-styled analysis of the error

Above we analytically showed that in a standard risk assessment setting the TCR method underestimates risk. We empirically demonstrate this in Figure 5 where the calibration curve (Figure 5a) shows that TCR underestimates risk for all predicted values. Figure 5b plots the squared error against true risk $\nu(V)$, illustrating that errors are extremely large for high-risk individuals, particularly for the TCR model. This highlights a danger in using confounded approaches like the TCR model: they make misleading predictions about the highest risk cases. In high-stakes settings like child welfare screening, this may result in dangerously deciding to *not* investigate the cases where the child is at high risk of adverse outcomes [10]. The counterfactually valid PL and DR models mitigate this to some effect, but future work should investigate why the errors are still large on high-risk cases and propose procedures to further mitigate this.

## D  Real-world experiment details and additional results

In this section we elaborate on the details of our evaluation of the methods on a real-world child welfare screening task.

### D.1  Child welfare dataset details

We use a dataset of over 30,000 calls to the child welfare hotline in Allegheny County, Pennsylvania. Each call contains more than 1000 features, including information on the allegations in the call as well as county records for all individuals associated with the call. The call features are categorical variables describing the allegation types and worker-assessed risk and danger ratings. The county records include demographic information such as age, race and gender as well as criminal justice, child welfare, and behavioral health history. The outcome we wish to predict is whether the family would be offered services if the case were screened in for investigation.

### D.2 Child welfare experimental details

We perform the first stage regressions using random forests to allow us to flexibly estimate the nuisance function $\pi$ and $\mu$. For the second stage regressions, we use LASSO to yield interpretable prediction models.

**Hyperparameters** The first stage random forest regressions use $500$ trees and the default *mtry* and splitting parameters in the `ranger` package in R. For our LASSO second stage regressions, we use cross-validation in the `glmnet` package in R to select the LASSO penalty parameters.

**Training runs** Each of the two nuisance function estimations in the first stage trains in one run. The LASSO cross-validation using `cv.glmnet` tunes over $\leq 100$ values of $\lambda$. Therefore, the TCR method trains in $\leq 100$ runs, the PL method with LASSO trains in $\leq 101$ runs, and the DR method with LASSO trains in $\leq 102$ runs.

**Sample size and error metrics** The dataset consists of 30,000 calls involving over 70,000 unique children. We partitioned the children into train and test partitions using a graph partitioning procedure that ensured that all siblings were contained within the same partition to avoid the contamination problem discussed in [9]. In order to enable more precise estimation of the counterfactual outcomes in this real-world setting, we perform a 1:2 train-test split such that the train split contains 27000 unique children and the test split contains 50000 unique children. We use the evaluation procedure in § 4 to obtain estimates of the MSE with confidence intervals.

**Computing infrastructure** All real-world experiments were run on a MacBook Pro with an 8-core i9 processor and 16 GB of memory. Each first stage regression trained in 15 seconds. Each second stage regression trained in 4.5 minutes.

### D.3 Modeling human decisions

Algorithmic tools used in decision support settings often estimate the likelihood of an event (outcome) under a proposed decision. This is the setting for which our method is tailored. By contrast, another paradigm trains algorithms to predict the human decision. We present here the results of such an algorithm when evaluated against the downstream outcome of interest (services offered). To train this model, we used the historical screening decision as the outcome. We allowed this model to access all confounders (both $V$ and $Z$, as if we did not have runtime confounding), yet this approach achieves a significantly higher MSE of $0.3207$ with $95\%$ confidence interval $(0.3143, 0.3271)$. It should not be surprising that a model trained on human decisions performs worse than models trained on downstream outcomes when we are evaluating against the downstream outcomes. This highlights the importance of using downstream outcomes in decision support settings when the goal is related to the downstream outcome e.g. to mitigate the risk of a downstream outcome or to prioritize cases that will benefit from the decision treatment.