[Reviews · NeurIPS 2020]

Review 1

Summary and Contributions: This paper addresses the problem of decision-making in the presence of runtime confounding, confounding due to current information not yet being available on some relevant variables, via counterfactual prediction. In particular, the notion of runtime confounding is formalized; a doubly-robust prediction method is proposed, as is a doubly-robust estimate of MSE that allows construction of confidence intervals from observed data. The motivating example comes from criminal justice where interest lies in predicting an offender's success on parole if placed on minimal supervision, while experiments rely on synthetic data. Of note is that the main goal is prediction rather than treatment effect estimates.

Strengths: Runtime confounding is an interesting problem that lends itself well to counterfactual reasoning based on a subset of confounders. The formulation of the problem and the theoretical results are sound and relevant for algorithm-assisted decision-making.

Weaknesses: In general I quite liked this paper, but not the motivating example. In the broader impact section the authors highlight that conditions 2.1 must hold and acknowledge that the training data needs to be unconfounded. However, I suspect this is unlikely in the case of criminal justice (for example, certain groups of offenders on parole may not be brought in for a small infraction and only given a warning, while others in a different demographic group may not be afforded such leniancies). This motivating example needs to be changed (see potential ethical concerns below) and is my main concern with the paper. The motivating example should not be one that suffers from the problems discussed in broader impact. Otherwise there is a false impression that the method is suitable for this example. The main result of the paper is Theorem 3.1, which is an estimate of the MSE that permits estimates with confidence intervals. However, the simulation study does not seem to evaluate this estimator and does not provide coverages for confidence intervals based on this doubly-robust estimator.

Correctness: Yes.

Clarity: The data generation for the simulation study is outlined without any intuition. The appendix adds these details, but more details are needed in the main paper, particularly in defining k_v and k_z more precisely. What is h in Algorithm 5?

Relation to Prior Work: Yes.

Reproducibility: Yes

Additional Feedback: line 154: "uses this to construct" to "uses this \mu to construct" line 175: "next result specialized" to "next result is specialized" line 217: "We expect the TCR..." to "Accordingly, we expect the TCR..." In this work, the algorithm is agnostic to which variables contribute to the runtime confounding. In practical situations, is there typically a core of variables that must be available before making predictions? Does this depend on how big the confounding effect of each runtime confounder is? Post-rebuttal: ------------------ First of all, l feel that the authors have identified an interesting practical problem and have come up with a logical approach to tackling it. However, the formal definition of runtime confounding still needs clarification. Even more important, is that the whole problem is embedded in contexts in which historical decisions likely suffer from systemic bias and in which training ignorability is likely violated (the child welfare setting likely suffers from same shortcomings as the parole board setting). If the problem was cast within a different framework/set of applications, this could be a useful tool for practitioners. But since it isn't and the final discussion of the broader implication cannot be seen before going to press, I have not changed my overall score.


Review 2

Summary and Contributions: EDIT: Thanks for your rebuttal. My score for this paper is unchanged, although I agree with other reviewers that 1) an additional experiment with real or semi-synthetic data would strengthen the paper quite a bit, and 2) the language around high-stakes decision-making, parole in particular, should be more careful. -------------------- - Gives estimators for run-time confounding – where all confounders are observed at train time, but not at test time - Gives bounds on error and a doubly robust estimator - Runs experiments on synthetic data and examines results of varying data generation hyperparameters

Strengths: - Claims seem theoretically sound, with proofs of most but not all claims in the appendix - Evaluation is thorough in synthetic data, varies parameters of data generation process and demonstrates how they connect to theoretical results - Contribution seems somewhat significant: it’s not an application I’ve thought of or seen described before as an open problem, but I can imagine it would be useful in some cases and the authors discuss this at the beginning of the paper - Relevant to Neurips community’s interest in causal inference + ML, and prediction under confounding

Weaknesses: - A couple of places where the work could be spelled out a little more thoroughly - It’s not 100% clear to me that this use case is plausible. But hard to say

Correctness: -- Yes this paper seems correct

Clarity: -- Paper is clear

Relation to Prior Work: -- Yes

Reproducibility: Yes

Additional Feedback: - Reproducibility: I think yes, - the code is provided but it is in R which I barely know. But it seems like it could be reproduced - Line 50: “it is common…” Is there a citation for this practice? I understand it’s common sense but given that it’s a central motivation for the paper could use some more grounding - Line 125: I think the positivity condition needs a “for all a” or something - Line 128: Would like to see one more line of explanation to unpack these stacked expectations - Line 132: are L and R functions? Clarify this condition. - Algorithm text throughout are extremely small and hard to read - Line 149: describing this decomposition in this way is nice - Line 173: is this the definition of oracle-efficient? Clarify the connection here - Line 176: would like to see this corollary 3.1 worked through in an appendix - Line 177: need to define what a k-sparse model is - Line 182: I don’t see why DR will outperform PL if d_v << d – the first term in the bound is the same and the second doesn’t include d_v? - Line 224: why do we see this effect with random forests? Is it due to lower model bias? Higher accuracy? - Line 229: a negative correlation exacerbating confounding doesn’t seem like a general result – clarify if this is dependent on the form of your data generating process - Line 236: unclear on why this inflection point happens with V_i and Z_i – need to explain more clearly how the data is generated in this way - Line 244: I like the motivation to test an interpretable second stage


Review 3

Summary and Contributions: The paper studies the problem of predicting the outcomes of interventions after learning from observational data. In this particular setting, unconfoundedness is satisfied at training time but not at runtime---all confounders are observed in data, but these may not be available when the predictions are made. The paper refers to this setting as "runtime confounding." The authors propose several algorithms for learning prediction models in this setting, based on approximating a consistent estimator of the causal effect using variables that are available at runtime. The algorithms are evaluated on synthetic data in settings with and without model misspecification. After rebuttal: Se comments under "weaknesses".

Strengths: - The problem is very relevant in settings where ML is deployed in practice. In some sense, it is a more general problem than the authors make it out to be---it is relevant also to more standard prediction problems, "without" causal components. - The proposed algorithm is simple but provably correct in the sense that it leads to consistent estimates of the sought-after quantity when sufficient data is available. - The paper is well organized and focused.

Weaknesses: - Despite giving plenty examples in the introduction of applications where "runtime confounding" is an issue, none of them are addressed in the empirical evaluation. It is true that unbiased evaluation of causal effects is difficult in the general setting, but it is common practice to synthesize either treatment assignments or outcomes based on real-world covariates, such as the commonly used IHDP benchmark. UPDATE: In the rebuttal, the authors have addressed this point and provided additional results which will strengthen the paper. - The problem is an instance of a more general problem which is not discussed (see Relation to prior work). UPDATE: In the rebuttal, the authors have addressed this point and provided additional discussion which will strengthen the paper. - I take issue with the idea that standard practice for this setting is "treatment-conditional regression", as defined by the authors. To me, this is a straw-man baseline. For the case where important confounders are known at training time, I have never heard of them being removed from analysis due to not being available at test time. I would expect that runtime imputation would be closer to standard practice, but this baseline is never discussed in the paper, nor used as a baseline. - As acknowledged by the authors, the plugin (and doubly-robust) approach(es) are "simple" but reasonable solutions to the problem at hand. (In fact, I would be surprised if the former is not already in wide-spread use). As such, I wish that the authors would include a discussion on the potential optimality of these solutions. Is a two-stage estimator optimally sample efficient or should we hope to find a better algorithm? Theorem 3.1 shines some light on this issue but is not compared to an alternative approach which could induce a different tradeoff. - The authors state that the biased baseline (TCR) is expected to perform best at low levels of confounding due to the (potentially) compounding error of the two-stage estimators. However, this error should be reduced when sample size increases, while the bias of TCR should remain. Regrettably, this is not confirmed by experiments. In light of this, the results in Figure 1 c) are a little odd as they show that all methods perform about as well. The prediction that "we expect this increase [in error] to be significantly larger for the TCR method that has confounding bias" appears to not be true. Is this an effect due to sample size or something else?

Correctness: - The results in the paper appear correct.

Clarity: - The paper is well presented and easy to read.

Relation to Prior Work: - The general problem of learning with access to data that is not available at runtime was formalized by Vapnik & Vashist (2009) as "Learning using privileged information" (LuPI). It seems to me that the current problem is an instance of this paradigm and a discussion of its relation is warranted.

Reproducibility: Yes

Additional Feedback: - The problem studied here is posed as a causal problem, but it is clearly an instance of a more general "coarsening" or "approximation" of prediction functions from a larger variable set to a smaller one. This is clear also from the algorithms given in the paper which first fit the function of interest using regression or doubly robust estimation, effectively handling the issue of causality. This parameter is then approximated as a function of the runtime variable set. I worry that the framing as a problem of causality may separate the literature unnecessarily. For example, is this not simply an example of predicting with missing variables at runtime? - The introduction gives a lot of examples of what the authors call "runtime confounding" but don't quite explain why it is an example of confounding. Certainly, the examples are relevant and important, but I'm not sure they are problems of confounding. The quantities necessary for prediction at runtime (i.e., the expected potential outcomes) are fully identified by the training data. Further, by definition, the outcome of the runtime instance is not observed, let alone used in the algorithm. So, what quantity is confounded at runtime? Based on this, I'm not sure that "runtime confounding" is an appropriate name.


Review 4

Summary and Contributions: This paper considers a specific kind of prediction task feeding into a human decision, where outcomes are a function of a (fully observed) set of covariates, but where some of these covariates—while observed—are off limits. This can be for a variety of reasons including timing, concern for bias, explainability, etc. Thus is it different from either straightforward tasks with no unobserved covariates, and from tasks where there are fully unobserved covariates. It then sets up a series of algorithms for accurate and efficient estimation of counterfactual predictions (eg, whether the decision is taken or not). The concrete example I had in my head as I was reading is the parole hearing setting the authors set up: prior data V can be used to make a prediction (eg on recidivism), and spoken testimony Z is collected at the time of the hearing. Z has signal both for predicting the decision (release/hold) and the outcome (recidivism, only observed if release). So omitting Z is a problem.

Strengths: I don’t know this literature very well, but the paper seems technically well done. The problem is an interesting and potentially a very important one, and the authors’ approach is rigorous and comprehensive. They note and deal with a number of concerns that are often unaddressed or unacknowledged: not just the presence/absence of unobserved covariates, but also the potential for the decision to affect the observed outcome (eg, medical treatments reduce the risk of outcomes, making the observed predictions too low relative to the counterfactual of interest, ie no treatment). Again, I am not the best person to evaluate the methods, but they seem convincing if you accept the basic “fully observed, but partially off limits” premise.

Weaknesses: I wonder if the paper would benefit from setting up a more concrete—even if hypothetical—example of when this scenario would occur. This seems important not just as a way to convince the reader of the need for this enterprise, but also to set up precise empirical tests showing that this approach works better than other, more naive methods with respect to decisions rather than prediction errors. I wasn’t sure what to make of the generality of the examples described in the introduction. The authors assert that the parole board wants the prediction before hearing the testimony, but they are going to hear the spoken testimony anyway. (i) One use of the prediction based on prior data is to determine who gets a hearing and thus who has Z collected. (ii) More generally, the prior prediction could influence not only the presence of Z, but how the decision maker views Z (eg, anchoring on a prior probability and not updating; or updating too much).

Correctness: Yes, as above.

Clarity: Yes. I would have liked the concrete examples to be continued to build intuitions, and perhaps more on the generality of the problem, but otherwise I thought it was easy to follow and clear.

Relation to Prior Work: Yes.

Reproducibility: Yes

Additional Feedback:

[Author Response · NeurIPS 2020]

We thank the reviewers for their valuable input on how to improve our manuscript. We are heartened by the general consensus on the importance of the problem and the theoretical grounding of our method. The reviewers rightly indicate a need for a stronger motivating example. We will retain the example of college advising (lines 20-22), but **replace** the parole example with one from child abuse hotline screening. Call workers must decide which allegations coming in to the child abuse hotline should be investigated. The worker relies on (immediate risk) information communicated during the call and an algorithmic risk score that summarizes (longer term) risk based on historical administrative data. The call is recorded but is not used as a predictor for three reasons: 1) inadequacy of existing case management software to run speech/NLP models on calls in realtime; 2) model interpretability; 3) need to maintain distinction between immediate risk (as may be conveyed during the call) and longer-term risk the model seeks to estimate. (We will discuss the parole example in the Broader Impact section to highlight the need for caution in applying the methodology. The criminal justice system is complex and involves multiple decision-making points, presenting opportunities for misuse.)

• Using the extra page permitted for the camera-ready, we will include an empirical example on real child welfare data. We use our evaluation procedure (§4) since we will not have ground-truth outcomes. MSE with $95\%$ confidence intervals (CI) for our DR method is $0.248(0.245, 0.250)$; PL: $0.249(0.246, 0.251)$; TCR: $0.265(0.262, 0.269)$. This suggests the PL/DR models could help workers better identify at-risk children.

• Our characterization of TCR as standard practice was informed by discussions with government agencies about their decision support systems. As R3 observes, one might consider feature imputation, but imputation may not desirable or feasible: e.g., using *imputed* protected attributes may still be impermissible; and the speech (call) data in the child welfare call is too high-dimensional to impute. Imputation using the predictors will also estimate the same biased target as TCR since $\mathbb{E}(Y^a \mid V) = \mathbb{E}(Y^a \mid V, f(V))$. The revision will provide this discussion with relevant citations.

• We would like to clarify that Theorem 3.1 describes the conditions under which our method is optimal. Theorem 3.1 decomposes the error of DR prediction method into the error of an oracle with with access to the true nuisance functions and a product of nuisance terms, which if small enough, imply that the DR method achieves the same error rate as an oracle and therefore inherits the optimality (such as minimax) of the oracle. We will elaborate on this in the revision and provide examples for clarity. Reviewers noted that while we observed the expected behavior for LASSO, random forests (RFs) showed TCR performing on par or better than PL and DR. For our sparse linear data, RFs have higher error than LASSO (compare Fig. 1a to 1c). The RF estimation error dominates the confounding error. This example shows that depending on the context and modeling choices, TCR may outperform the counterfactually valid approaches. Our evaluation procedure (Alg. 5) can assess this for a given setting. We will add this to the discussion in lines 222-226.

**R1** We thank R1 for the useful suggestions. For the MSE estimator proposed in §4, we provided empirical results in §C.3 that assessed how often it identifies the best model. The revision will include coverage results: On 100 simulations for a test size of $1000$, the MSE estimator with $95\%$ CI covered the true MSE 94 times for the DR approach and 93 times for the PL. We will expand Broader Impact to include examples that show when this method should *not* be used.

**R2** We appreciate R2's detailed comments and will make all clarifications suggested. R2 raises a good question about our interpretation of Corollary 3.1. For a simple example, consider the case where $k_\nu \approx k_\mu \approx k_\pi$. When $d_v \ll d$, the second term of the PL bound dominates the error whereas the first term of the DR bound dominates in high-dimensional settings. The revision will clarify this. We thank R2 for noting the comprehensiveness of our empirical analysis.

**R3** We thank R3 for noting that our proposed approach is more general than our articulated motivation suggests. The revision will describe how our approach is applicable to settings such as selective labels. While selection bias literature such as survey inference use doubly-robust approaches, existing theory does not cover the prediction setting. We hope that our theoretical contributions help fill this gap. R3 aptly identifies the connection between causal inference and missing data. Indeed the fundamental problem of causal inference is one of missing data: that we only observe one potential outcome (Holland 1986). In our setting, we have additional missingness in the features available for prediction, features which crucially affect the decisions and hence are confounders. We believe the missing confouders problem merits special attention. For the case in which the missing features only affect the outcome, with infinite data regressing the outcome on the available predictors is optimal. This is not true for our setting, where the confounding bias persists (Prop 3.2). We will include this discussion in a paragraph that relates our work to privileged learning.

We were glad to see R3's interest in the optimality of our method and hope our discussion above is useful. We used simulations to explore whether our method performs well without sample-splitting, using the **full** training sample (lines 202-206). The methods performed as expected for LASSO, even though our theory as presented does not cover this setting. Theory that does not rely on sample-splitting typically requires strong empirical process assumptions. We would like to clarify that the experiments held sample size fixed (line 211).

**R4** We thank R4 for finding our contributions valuable and for suggesting a comparison to a model of decisions. A model of child welfare screening decisions achieves an estimated MSE for the adverse outcome of $0.356(0.353, 0.359)$. We will include this comparison to our method (see above) to demonstrate the value in predicting downstream outcomes.

[Meta-Review · NeurIPS 2020]

AC's comments before receiving the ethics review (see below): The authors had a lively and detailed discussion and settled on the following points: - The framing of the paper seems reasonable: it seems plausible that one may have certain variables in past data that may not exist, or be allowed during prediction time. There's a similar line of work on this in algorithmic fairness which is very plausible given GDPR, companies not wanting to be sued for violating laws, or even HIPAA/security issues. - They agree that the authors should run the baseline suggested by R3: trying to impute the missing values and see how well that does. They do buy the authors argument that their approach will work better because imputation is a strictly harder problem. - They agree that the setting of runtime confounding is not such a technically difficult problem as you do have confounders in the training data. However the realism of the setting makes them think this is still a useful problem to address, even if it is much simpler than other causal settings. I urge the authors to modify the paper according to the suggestions of reviewers. I vote to accept. AC's comments after receiving the ethics review: After considering the ethical review and looking over the paper again, I’ve decided that the paper should still be accepted for the following reason: The ethical review’s main concerns were that the current paper doesn’t sufficiently engage with current literature on fairness in ML. I do agree that there are a number of methods in the fairness community for addressing the issue of not having sensitive data during test time, this includes algorithms that are specially created for this purpose and methods that only have third parties handle sensitive attributes or that encrypt sensitive attributes: Agarwal, A., Beygelzimer, A., Dudik, M., Langford, J.,and Wallach, H. "A reductions approach to fair classification". ICML 2018 Jagielski, Matthew, Michael Kearns, Jieming Mao, Alina Oprea, Aaron Roth, Saeed Sharifi-Malvajerdi, and Jonathan Ullman. "Differentially private fair learning.” ICML, 2019. Veale, M. and Binns, R. "Fairer machine learning in the real world: Mitigating discrimination without collecting sensitive data”. Big Data & Society, 4(2), 2017. Kilbertus, Niki, Adria Gascon, Matt Kusner, Michael Veale, Krishna Gummadi, and Adrian Weller. "Blind Justice: Fairness with Encrypted Sensitive Attributes.” ICML, 2018. while these works only target supervised learning and this work targets a causal quantity, this paper just requires a set of regressions so in principle the above approaches could apply. However, modifying them to fit this setting would be non-trivial. I think if the authors added a discussion on the relationship to this line of work in fairness it would be enough. Further, this approaches applies beyond fairness, to any case of observed confounding during training and unobserved confounding during testing, and I think the benefit of introducing the setting of “runtime confounding” to the ML community outweighs the missing references and discussion, which can easily be added. ----------- ADDITIONAL REVIEW FROM ETHICS EXPERT -------------- - What is your recommendation with regard to the broader impact statement? Are any revisions needed? I am not comfortable in accepting the paper. The paper position itself in contexts in which historical decisions likely suffer from biases, i.e. health care, education, lending, criminal justice, and child welfare. Indeed, these are the main settings considered also in the ML fairness literature. One of the example mentioned in the introduction to justify the goal of dealing with unavailable information at evaluation time is in fact about fairness: The authors say that 'runtime confounding arises when historical decisions and outcomes have been affected by sensitive or protected attributes which for legal or ethical reasons are deemed ineligible as inputs to algorithmic predictions. We may for instance be concerned that parole boards implicitly relied on race in their decisions, but it would not be permissible to include race as a model input.' Thus I was expecting this to be an ML fairness paper, yet the paper is only superficially discussing the fairness aspect in the broader impact statement. I would have liked the paper to contain, already in the introduction, a discussion on (ML) fairness, and later a detailed explanation on what training ignorability means in unfairness terms. Regarding the parole boards example, I have concerns about that implications of ignoring fairness aspects and the ML fairness literature. If the use of race by parole boards is deemed unfair, then the use of the proposed method would not resolve this ethical issue, but only legal requirements. On the other hand, there are methods in the ML fairness literature (also pointed out by the meta-reviewer) that can do both. The question then is also, beyond the issue that the unfairness problem is not addressed by the proposed method, why are these methods from the ML fairness literature not at least discussed in the related work section (if not compared)? I am not objecting that there is no merit in the proposed method wrt dealing with missing information. But given the paper's strong framing in sensitive contexts, a detailed discussion on ML fairness is in my opinion necessary, and perhaps a comparison given that the proposed method would not be appropriate in the fairness cases discussed in the paper. I would like to see the discussion before publication, as some of it might be non-trivial, for example the translation of the ignorabiliy assumptions into fairness assumptions. - Would the publication of the research potentially bring with it undue risk of harm? Please discuss any suggested mitigations or required changes. The framing of the paper is poor. The paper positions itself in settings that suffer from fairness issues and that are considered in ML fairness without a reasonable discussion about it. The proposed method would mostly not be appropriate in such settings, whilst there are methods in the ML literature that are. This should be clarified. It seem to me as the reviewers are not clear about what the ignorabiliy assumption would mean in terms of fairness for the relevant settings (it would not be possible to give one exact formula but the instruments for understanding how to make the translation should be given). The readers and practitioners will be even more confused. I am of the opinion that the paper need to be rewritten substantially into a more informed paper wrt ML fairness before publication.